# Plasma Cholesterol- and Body Fat-Lowering Effects of Chicken Protein Hydrolysate and Oil in High-Fat Fed Male Wistar Rats

**DOI:** 10.3390/nu14245364

**Published:** 2022-12-16

**Authors:** Thomas A. Aloysius, Veronika Tillander, Matteo Pedrelli, Simon N. Dankel, Rolf K. Berge, Bodil Bjørndal

**Affiliations:** 1Department of Clinical Science, University of Bergen, N-5020 Bergen, Norway; 2Department of Laboratory Medicine, ANA Futura, Karolinska Institutet, 141 52 Huddinge, Sweden; 3Medicine Unit Endocrinology, Karolinska University Hospital, 141 52 Stockholm, Sweden; 4Department of Heart Disease, Haukeland University Hospital, N-5021 Bergen, Norway; 5Department of Sports, Food and Natural Sciences, Western Norway University of Applied Sciences, N-5020 Bergen, Norway

**Keywords:** protein hydrolysate, bioactive peptides, dietary protein source, lipid metabolism, high-fat diet, western diet

## Abstract

Rest raw materials provide a new source of bioactive dietary ingredients, and this study aimed to determine the health effects of diets with chicken protein hydrolysate (CPH) and chicken oil (CO) generated from deboned chicken meat. Male Wistar rats (*n* = 56) were divided into seven groups in three predefined sub-experiments to study the effects of protein source (casein, chicken fillet, pork fillet, and CPH), the dose-effect of CPH (50% and 100% CPH), and the effects of combining CPH and CO. Rats were fed high-fat diets for 12 weeks, and casein and chicken fillet were used as controls in all sub-experiments. While casein, chicken-, or pork fillet diets resulted in similar weight gain and plasma lipid levels, the CPH diet reduced plasma total cholesterol. This effect was dose dependent and accompanied with the reduced hepatic activities of acetyl-CoA carboxylase and fatty acid synthase. Further, rats fed combined CPH and CO showed lower weight gain, and higher hepatic mitochondrial fatty acid oxidation, plasma L-carnitine, short-chain acylcarnitines, TMAO, and acetylcarnitine/palmitoylcarnitine. Thus, in male Wistar rats, CPH and CO lowered plasma cholesterol and increased hepatic fatty acid oxidation compared to whole protein diets, pointing to potential health-beneficial bioactive properties of these processed chicken rest raw materials.

## 1. Introduction

Protein is important for good health and the primary role of dietary protein is to provide the body with essential amino acids for protein synthesis and metabolism. The Dietary Reference Intake (DRI) for protein is 0.8 g per kg body weight. Evidence supports a higher protein intake for weight loss and weight loss-associated health benefits [1]. The source, as well as the amount of protein, could be important for human health. Ingested proteins have a wide range of biological functions affecting glucose and fatty acid metabolism, immunological function, and blood pressure, that may be linked to bioactive peptides released in the gastrointestinal tract [2]. Moreover, it has been suggested that different protein sources have different obesogenic potential. For example, casein protein was able to fully prevent high-fat, high-protein diet-induced obesity, whereas feed based on protein sources from pork and chicken increased the accretion of adipose tissue mass in male mice [3].

Health benefits from fish consumption have been attributed partly to long-chain *n*-3 polyunsaturated fatty acids (LC *n*-3 PUFA) [4,5,6,7,8,9,10]. Recent studies indicate that bioactive peptides released during digestion could potentially contribute to the beneficial effects of fish consumption [2,11,12,13,14]. We have previously reported that a salmon protein hydrolysate had a protective role in atherosclerotic development in a mouse model through mechanisms linked to the inhibition of inflammation [15]. Moreover, we have demonstrated that fish protein hydrolysates influence lipogenesis, hepatic fatty acid metabolism, and fatty acid composition in rodents, indicating that effects on fatty acid metabolism are important for the bioactivity of protein hydrolysates [16,17,18]. Similarly, protein from vegetable sources such as soy, pea, and lupin, have shown both TAG- and cholesterol-lowering effects in animals and humans [19,20,21,22], and soy milk and its hydrolysate were found to be anti-obesogenic in mice compared to casein [19]. Recently, we reported that chicken protein hydrolysates (CPHs) had anti-inflammatory effects, and that CPH stimulated mitochondrial fatty acid oxidation and lowered atherosclerosis in mice [23]. In addition, a chicken collagen hydrolysate was shown to exert strong angiotensin-converting enzyme (ACE)-inhibitory activity and anti-hypertensive effects as well as to protect from cardiovascular damage [24,25].

We have previously demonstrated that a fish protein hydrolysate influenced hepatic fatty acid composition and increased plasma L-carnitine levels in a mouse model of chronic inflammation [16]. L-carnitine is involved in the generation of metabolic energy from long-chain fatty acids by mediating their transport across the mitochondrial membrane in the form of acylcarnitines, and plasma acylcarnitine levels are suggested as markers of metabolic activity [26,27]. Elevated plasma acylcarnitines have been linked to disease progression both in cardiovascular disease [28,29,30] and conditions involving insulin resistance [31,32,33]. In recent years, the gut microbiota metabolism of L-carnitine has become a topic in several studies [34,35,36], and dietary L-carnitine consumption has been reported to result in trimethylamine (TMA) release via the gut microbiota [37], which is then converted into trimethylamine-*N*-oxide (TMAO) by hepatic flavin-containing monooxygenase 3 (FMO) [38]. Importantly, TMAO has been suggested as a prognostic marker for non-communicable diseases [39].

A diet rich in mono- and polyunsaturated fats, may have protecting effects against chronic diseases such as diabetes type 2 and cardiovascular disease [40,41,42,43], although the beneficial effects regarding monounsaturated fats are still contradictive [44,45]. One of the most important characteristics of chicken oil is the presence of a high content of monounsaturated and polyunsaturated fatty acids, reflected by high levels of oleic and linoleic acid [46]. However, while there is substantial potential in the broiler industry to exploit protein material from chicken carcass remains for human consumption, more research is needed to determine if CPH or oil has unique health properties that could motivate such exploitation. In the present study, the aim was to investigate the metabolic effects of CPH and oil from chicken in high-fat fed male Wistar rats.

## 2. Materials and Methods

### 2.1. Animals and Diets

The animal study was conducted according to the Guidelines for the Care and Use of Experimental Animals, and in accordance with the ARRIVE guidelines and Norwegian legislation and regulations governing experiments using live animals. The Norwegian State Board of Biological Experiments with Living Animals approved the protocol (Permit number 2015-7367). All efforts were made to optimize the animal environment and avoid suffering. Male Wistar rats, 4–5 weeks old, were purchased from Janvier Labs. Upon arrival the rats were randomized using Research Randomizer (https://www.randomizer.org, accessed on 12 June 2017), labeled and placed in open cages, four in each cage, and allowed to acclimatize to their surroundings for one week. During the acclimatization, the rats had unrestricted access to chow and tap water. The rats were kept in a 12 h light/dark cycle at a constant temperature (22 ± 2 °C) and a relative humidity of 55% (±5%). Upon start of the experiment, the rats were block randomized to 7 dietary intervention groups (*n* = 8) [47], and divided into three sub-experiments with some common groups (casein, chicken, and CPH) to reduce the amount of animals used in the experiments according to the 3Rs. Male rats were used as they are more susceptible to dietary metabolic effects compared to female rats [48].

Rats were fed high-fat (HF) diets with 47 E% from fat, 32 E% from carbohydrate and 21 E% from protein (Table 1). The first sub-experiment was designed to study at the effect of different protein sources, and consisted of 4 groups comprised of: control HF diets with 100% casein as protein source (casein), or 100% protein from pork fillet (pork), chicken fillet (chicken), or chicken protein hydrolysate (CPH). The second sub-experiment was designed to study the dose effects of the CPH and consisted of 4 HF-groups including the casein diet, chicken diet, CPH diet and a diet group with 50% protein from CPH and 50% from casein (CPH-50%). The third sub-experiment was designed to study at the effect of chicken oil in combination with CPH, and consisted of 5 HF-groups including the casein diet, chicken protein diet, CPH diet, a diet where chicken oil constituting 40% of the fat source replacing lard (chicken oil), and finally 100% CPH in combination with 40% chicken oil (CPH-chicken oil). The CPH was produced as follows: Rest raw materials were generated by mechanical deboning of the remaining chicken meat after filleting (Nortura AS Hærland, Eidsberg, Norway), and was used for the hydrolysis experiments. Chicken meat from deboned chicken was treated enzymatically with food grade Corolase^®^ PP (AB Enzymes GmbH, Darmstadt, Germany) and then filtered, using micro ultrafiltration, as previously described [23], but the CPH was not dried. More than 50% of the final preparation consisted of peptides in the range 200–1200 Da. The chicken oil was a generous gift fra Nofima AS (Bergen, Norway), and the fatty acid composition is given in Table 2. Chicken breast fillet and pork tenderloin fillet was minced, microwaved, and mixed into the diet. The final E% was confirmed by analysis (Nofima AS) and was similar in all diet groups (Table 3). The amino acid composition of the diets is given in Table 3 (see procedure below). Diets were packed airtight and stored at −20 °C until use to prevent lipid oxidation. Fresh feed was provided every 1–2 days.

Rats in all groups were pair-fed (based on casein control cage feed intake) for 12 weeks, and all diets had similar palatability based on the feed intake of the rats. All animals were weighed weekly and feed intake was determined every 1–2 days. At sacrifice, rats were anaesthetized by inhalation of 5% isoflurane (Schering-Plough, Kent, UK). EDTA-blood was collected by cardiac puncture and immediately placed on ice. The samples were centrifuged at 2000× *g* for 15 min and plasma was stored at −80 °C prior to analysis. Liver and four different white adipose tissues (WAT) were collected and weighed: subcutaneous WAT was collected on the right side of the body from fore- to hindlimb. Epididymal and perirenal WAT was collected on the right side of the animal, and mesenteric WAT surrounding the intestines was collected. Fresh samples from liver were prepared for mitochondrial β-oxidation analysis (see below). The remaining parts of the liver were immediately snap-frozen in liquid nitrogen and stored at −80 °C until further analysis.

### 2.2. Amino Acids in the Diet

The amino acids in the diets were determined after hydrolysis in 6 M HCL at 110 °C for 22 h and pre-derivatisation with phenylisothiocyanate according to the method of Choen and Strydon [49]. To analyse the amounts of free amino acids in the diets, the samples were extracted and deproteinated by the addition of two volumes of 5% sulphosalisalylic acid, kept on ice for 30 min and centrifuged at 5000× *g* for 15 min. The supernatant was mixed with internal standard, norleucine. The amino acids were quantified using a Biochrom 20 Plus Amino Acids Analyzer (Amersham Pharmacia Biotech, Uppsala, Sweden).

### 2.3. Fatty Acid Composition of the Chicken Oil

The chicken oil was trans-esterified using BF_3_/methanol [50]. Extracts of fatty acyl methyl esters (FAME) were heated in 0.5 mol L^−1^ KOH in ethanol/water solution (9:1) [51], to remove neutral sterols and non-saponifiable material. Recovered FAs were re-esterified using BF_3_/methanol. The methyl esters were quantified by gas chromatography as previously described [18].

### 2.4. Quantification of Plasma Lipids and Carnitine Metabolites

Lipids from plasma were measured enzymatically on a Hitachi 917 system (Roche Diagnostics GmbH, Mannheim, Germany) using the cholesterol (CHOD-PAP) and TAG (GPO-PAP) kit from Roche Diagnostics, and the free cholesterol (Free Cholesterol FS), non-esterified fatty acid (NEFA FS,) and phospholipid kit (Phospholipids FS) from DiaSys (Diagnostic Systems GmbH, Holzheim, Germany). Lipoproteins were also separated from 2.5 μL of individual plasma samples by size exclusion chromatography (SEC), using a Superose 6 PC 3.2/300 column (GE Healthcare Bio-Sciences AB, Uppsala, Sweden). Lipoproteins were eluted as a fraction appearing in the exclusion volume of the sepharose column that contained chylomicrons (if present) together with VLDL, then LDL and last HDL. Total cholesterol was calculated after integration of the AUC in the individual chromatograms [52,53,54], generated by the enzymatic-colorimetric reaction Cholesterol CHOD-PAP (Roche Diagnostics).

L-carnitine, trimethyllysine, γ-butyrobetaine, acetylcarnitine, propionylcarnitine, valerylcarnitine, octanoylcarnitine, lauroylcarniitne, myristoylcarnitine, palmitoylcarnitine, betaine, choline and TMAO were analyzed in plasma by HPLC-MS/MS as described by Vernez et al. [55] with some modifications [56].

### 2.5. Mitochondrial Fatty Acid Oxidation and Hepatic Enzyme Activities

Post-nuclear fractions from liver was prepared as previously described [57]. Palmitoyl-CoA oxidation was measured in the post-nuclear fraction from fresh liver as acid-soluble products, as described by [58]. The following enzyme activities were measured in post-nuclear fraction from frozen liver tissue: Carnitine O-palmitoyltransferase 2 (CPT2, EC: 2.3.1.21) [59], mitochondrial HMG-CoA synthase (HMGCS, EC: 2.3.3.10) [60], fatty acid synthase (FASN, EC: 2.3.1.85) [61], acetyl-CoA carboxylase (ACC1, EC: 6.4.1.2) [62], glycerol-3-phosphate acyltransferase (GPAT, EC: 2.3.1.15) [63,64] and acyl-CoA oxidase [65,66]. The amount of protein was measured using the DC protein assay kit (Bio-Rad Laboratories, Hercules, CA, USA).

### 2.6. Plasma Inflammatory Markers and Antioxidant Capacity

Plasma concentrations of interleukin 1β (IL-1β), IL-2, IL-6, granulocyte-macrophage colony- stimulating factor (GM-CSF), interferon gamma (IFN-γ), monocyte chemotactic protein-1 (MCP-1) and tumor necrosis factor-α (TNF-α) were determined using a custom-made multiplex MILLIPLEX MAP kit (Millipore Corp., St. Charles, IL, USA) according to the manufacturer’s protocol. The concentration of each marker in the solution was determined using a Bio-Plex 200 instrument, with Bio-Plex Manager Software version 4.1 (Bio-Rad, Hercules, CA, USA).

Total antioxidant capacity in plasma was measured using a kit from Abcam (Cambridge, UK) according to the manufacturer’s instructions for analysis of both small molecule antioxidants and protein’s ability to reduce Cu^2+^ to Cu^+^. Absorbance was measured at 570 nm using a plate reader, and results were expressed as trolox equivalents according to a trolox standard curve.

### 2.7. Statistical Analysis

Data were analysed using the Prism Software (Graph-Pad Software, San Diego, CA) to determine statistical significance. The results are shown as mean with standard deviation (SD) of 8 rats per group. One-way ANOVA was used to evaluate statistical significance (*p* > 0.05) within the predesigned sub-experiments 1, 2, and 3. Tukey’s multiple comparisons test was used to evaluate statistical differences between all diet groups in each sub-experiment. Pearson’s correlation coefficients were used when comparing two independent variables. *p*-values < 0.05 were considered statistically significant.

## 3. Results

### 3.1. The Effect of Different Protein Sources on Body Weight, Adipose Tissue and Liver Mass

Male Wistar rats fed high-fat diets with 100% protein from pork or chicken meat showed similar body weights throughout the 12 weeks feeding period, compared to control rats fed 100% casein as a protein source (Figure 1A). Total weight gain, total feed intake, and feed efficiency by pork and chicken proteins were also similar compared to casein (Figure 1B–D). Both casein and chicken protein were used as control groups, as references to the standard protein source in rodent diet experiments, and unhydrolyzed chicken protein, respectively. Using CPH as the sole protein source resulted in a lower body weight compared to casein, throughout the study (Figure 1A), but the total weight gain was not significantly altered compared to either casein or chicken whole protein (Figure 1B, *p* = 0.16 or 0.31).

In line with a constant calculated dietary E% in the diets (Table 3) and a similar feed intake (Figure 1C), the replacement of casein with pork and chicken whole protein did not change the subcutaneous fat weight or the total visceral fat weight (Figure 1E,F). Moreover, the visceral/subcutaneous white adipose tissue (WAT) ratio and liver weight was not changed by the different unhydrolyzed protein sources. In contrast, the CPH diet reduced total visceral fat and subcutaneous fat significantly compared to unhydrolyzed chicken protein (*p* = 0.024 and *p* = 0.029, respectively), with a similar tendency compared to casein (*p* = 0.17 and 0.13, respectively) (Figure 1E,F). No significant change was observed in the visceral/subcutaneous WAT ratio or the liver weight.

### 3.2. Lowering of Cholesterol, Cholesterol Esters and Hepatic Lipogenesis by Different Doses of Chicken Protein Hydrolysate

Interestingly, compared to casein or chicken protein, two different doses of CPH feeding resulted in a decrease in plasma total cholesterol in rats fed a high-fat diet (Figure 2A). Free cholesterol was significant lowered in rats fed CPH compared to casein (Figure 2B) and cholesteryl esters were significant lowered in both rats fed 50% and 100% CPH compared to casein. Compared to chicken, rats fed the CPH demonstrated significantly lower plasma total cholesterol and cholesterol esters levels (Figure 2A,C). To further investigate these changes, we looked at the distribution of cholesterol within the different lipoprotein particles. No significant changes were observed in cholesterol carried by the large lipoprotein particles (mostly VLDL and remnant particles) in rats fed the two hydrolysate groups (Figure 2D), when comparing to casein fed rats. Chicken meat diet increased the content of cholesterol compared to the other groups in large lipoprotein particles (Figure 2D,E). A reduction in cholesterol carried in the smaller sized lipoprotein particles were clearly observed in the CPH fed groups compared to the casein group (Figure 2D,F). Interestingly, in the CPH group, a shift in the retention time of the peak corresponding to smaller size particle (HDL) was noted (Figure 2D).

These CPH-dependent changes in lipids were accompanied with a decreased hepatic in vivo lipogeneses, as the activity of fatty acid synthase (FAS), and was reduced by both doses of CPH compared to the chicken fillet diet, and the activity of acetyl-CoA carboxylase (ACC) was significantly reduced by the 100% CPH dose compared to the casein diet (Figure 3A,B). Altogether, changes in plasma cholesterol components as well as the hepatic lipogenesis was unique to hydrolyzed protein and not intact proteins. On the other hand, CPH and unhydrolyzed protein did not differentially affect plasma triacylglycerol (TAG), non-esterified fatty acid (NEFA) and phospholipids, nor hepatic mitochondrial fatty acid oxidation in the absence or presence of malonyl-CoA, CPT2 activity, HMG-CoA synthase activity, HMG-CoA reductase activity, peroxisomal ACOX activity or TAG biosynthesis measured as GPAT activity (Appendix A).

### 3.3. The Effect of Chicken Protein Hydrolysate on Systemic Inflammation and Total Antioxidant Status

CPH feeding did not significantly influence plasma antioxidant capacity (AOC) compared to casein, pork or chicken protein (Figure 4A). When comparing casein, chicken protein, and CPH-fed groups, chicken protein tended to increase the plasma IL-2 level compared to casein (*p* = 0.056), while CPH tended to reduce IL-1β, IL-6, IFN-γ, G-CSF GM-CSF and TNFα compared to whole protein from chicken (*p* = 0.08–0.21) (Figure 4B–H).

### 3.4. Effects on Weight and Plasma Cholesterol in Rats Fed a Combination of CPH and Chicken Oil

As the diet with chicken meat and CPH also contained some residual chicken oil, we wished to identify the effect of chicken oil alone and in combination with CPH. While chicken oil alone and CPH alone had no effect on body weight compared to casein or chicken protein, a combination of CPH and chicken oil led to significantly lower body weight gain compared to casein and chicken protein despite a similar feed intake (Figure 5A,B). In addition, the weight gain and weight of the visceral adipose tissue was lower in rats fed CPH or a combination of CPH and chicken oil compared to chicken oil alone (Figure 5A,C). Moreover, total plasma cholesterol and cholesterol ester were significantly reduced with CPH alone and the combined treatment compared to both casein, chicken protein, and chicken oil (Figure 5D,E).

### 3.5. Effects on Plasma Carnitine and Acylcarnitines, TMAO and Precursors, and Hepatic Lipid Metabolism in Rats Fed a Combination of CPH and Oil

The chicken oil contained a high amount of the potentially beneficial fatty acid C18:1n-9 (Table 2), but a high-fat diet with 40% of chicken oil had no effect on plasma lipid components compared to the casein and chicken protein control groups (Table 4). Moreover, the hepatic enzyme activities were similar, except for the CPT2 activity, which was significantly higher (Figure 6). Interestingly, a diet with both chicken oil and CPH increased the hepatic mitochondrial fatty acid oxidation compared to casein, accompanied with the increased activity of CPT2 both compared to casein and chicken protein, and a tendency to the increased activity of HMG-CoA synthase (*p* = 0.18 vs. casein; Figure 6A–C). The activity of peroxisomal ACOX (Figure 6D) was not different, and while the CPH diet significantly reduced FAS and ACC activity (Figure 4), the combination diet only tended to reduce the activity of enzymes involved in lipogenesis (Figure 6E,F).

Interestingly, rats fed a high-fat diet with CPH and CPH combined with chicken oil resulted in significantly higher total plasma levels of carnitines and TMAO compared with the casein group (Table 5), and this was accompanied with an increased plasma level of betaine, but not choline. The total carnitines were mostly attributed to increased L-carnitine and short-chain ACs, including acetylcarnitine (AC2) and propionylcarnitine (AC3), but not medium- (AC5, AC8) or long-chain ACs (AC12, AC14, AC16) (Table 5). Thus, the plasma AC2/AC16 was increased in the CPH group and in the combination group of chicken oil and CPH compared to casein (Table 5). Interestingly, a positive correlation was found between the AC2/AC16 ratio and the rate of mitochondrial fatty acid oxidation in all samples (Figure 6G), thus a high ratio may potentially reflect more complete fatty acid oxidation. Noteworthily, levels of the carnitine precursor γ-butyrobetaine were higher after CPH feeding and combined CPH and chicken oil diets compared to casein, whereas plasma trimethyllysine levels were similar (Table 5). Importantly, the effects on carnitines were observed both with hydrolysed and unhydrolyzed chicken protein, as L-carnitine, total carnitines and the AC2/AC16 ratio was increased in the chicken protein group compared to casein (Table 5). Finally, a significant negative correlation between rat weight gain and hepatic β-oxidation was observed (Figure 6H).

## 4. Discussion

It has previously been reported that a high-protein diet (40 g protein/100 g diet) with casein as the protein source could prevent high-fat diet induced obesity in mice, while beef, pork, or chicken to a varying degree resulted in an increased accretion of adipose tissue mass, despite a similar energy intake [3]. In the rat experiment presented here, diets based on pork and chicken fillet did not differentially influence body weight or adipose tissue mass accretion compared to casein pair-fed male rats. Moreover, there were no differences in feed efficiency, plasma lipids, liver indexes, or hepatic fatty acid metabolizing enzyme activities, including mitochondrial and peroxisomal fatty acid oxidation, when pork and chicken protein was used as the major protein source. Thus, different results are obtained by pork and chicken proteins in male mice and rats, although it is also possible that a higher amount of protein in the diet is necessary to reveal a difference in the metabolic effects of unhydrolyzed protein sources.

Exchanging these whole protein sources with a crude hydrolysate from chicken protein did not significantly influence weight gain but reduced adipose tissue weight. Additionally, compared to both unhydrolyzed chicken protein from white chicken meat and casein, we found that two different doses of CPH had cholesterol and cholesterol ester lowering effects in male rats (Figure 2) and possibly changed the cholesterol distribution in the lipoprotein particles. This is in line with studies demonstrating that isolated peptides or protein hydrolysates can have diverse bioactive properties [67,68,69,70]. CPH generated using the same enzymatic method was previously demonstrated to reduce plasma cytokines in a mouse model of diet-induced obesity [23] and to prevent plaque formation in a female mouse atherosclerosis model, without reducing plasma cholesterol [71]. Thus, the cholesterol and especially cholesterol ester lowering observed in male rats could be a species-specific effect, in line with previous studies on hydrolysates from animal sources [18,72,73,74]. It was of interest that the hepatic enzyme activities of acetyl-CoA carboxylase and FAS were reduced by CPH indicating that the in vivo fatty acid biosynthesis was reduced. Both cholesterol biosynthesis and lipogenesis are regulated by the same family of transcription factors, the sterol regulatory element binding protein (SREBPs) [75]. Whether the reduced lipogenesis is linked to decreased plasma cholesterol esters should be considered. Thus, our findings indicate that CPH has potential to protect against hypercholesterolemia and to reduce hepatic lipogenesis in male rats. Further studies are needed to determine if the effect is similar in female rats. As oestrogen partly protects from dietary induced metabolic disturbances in rats [48], effects in females could potentially be less prominent.

We further looked at the effect of chicken oil as a dietary lipid source, and a significantly lower body weight gain was observed in the combination feeding of chicken oil and CPH, but not in male rats fed CPH or chicken oil alone (Figure 5). This was accompanied by a significantly increased hepatic β-oxidation specifically in the combination group, in addition to cholesterol- and cholesterol ester-lowering as also observed with CPH alone. As CPH produced by Corolase PP digestion has previously been shown to stimulate hepatic β-oxidation in mice [23], the increase in hepatic β-oxidation in the combination group could be caused by a dual effect generated by fatty acids in CO and bioactive peptides in CPH. Changes in hepatic β-oxidation could influence weight gain, and a moderate but significant negative correlation between these factors was observed (Figure 6G). Altogether, additive hepatic metabolic changes on enzymes involved in both lipogenesis and fatty acid oxidation could have contributed to the reduced body weight gain in male rats fed the combination diet compared to controls. Although the increased hepatic activities of CPT2, reduced lipogenesis [72] and increased mitochondrial fatty acid oxidation [76] have previously been associated with TAG lowering, we did not observe differences in plasma or hepatic TAG in the CPH-group or the combined chicken oil and CPH-group.

An important finding of the current study is that the hepatic fatty acid metabolism activity seemed to be reflected in plasma by changes in levels of carnitines, TMAO and their precursors. CPH alone and in combination with chicken oil significantly increased plasma levels of γ-butyrobetaine, betaine, total carnitines and TMAO, but not trimethyllysine and choline. The levels of circulating TMAO are affected by several factors, which include kidney function, diet, protein transport [77], and the gut microbiota [37]. A recent study showed that TMAO is a prognostic biomarker of kidney function [78]. Additionally, TMAO has been implicated as a risk factor for human CVD [28,29,30]. Nonetheless, the consumption of fish and krill, which are high in TMAO, has long been associated with reduced CVD risk [79]. Other clinical trials also indicate that diets enriched in the TMAO precursor carnitine are associated with beneficial effects on CVD [80]. In this study, we found increased plasma TMAO levels and decreased plasma cholesterol and cholesterol-esters in rats fed a CPH-diet. Interestingly, it has been reported that the TMAO-generating enzyme flavin-containing monooxygenase 3 is a central regulator of cholesterol balance [81], suggesting a mechanism by which dietary factors such as CPH modulate cholesterol metabolism in association with altered TMAO levels.

The increased plasma levels of plasma carnitines in the whole protein chicken group, CPH group and the group combining CPH and chicken oil were attributed to increased levels of L-carnitine, AC2 and AC3, whereas branched-chain acylcarnitine (AC5), medium-chain acylcarnitine (AC8) and long-chain acylcarnitines (AC12, AC14, AC16) were similar to the casein-fed group (Table 5). Thus, the ratio between AC2 and long-chain acylcarnitines increased. A low plasma AC2/(AC16 + AC18:1) ratio has been suggested as a marker for defect mitochondrial fatty acid oxidation and is associated with all-cause mortality in carnitine-deficient patients [82]. We did not measure AC18:1, but it was of interest that the plasma AC2/AC16 ratio was increased in the whole protein chicken group, CPH group and in the combination group of chicken oil and CPH compared to casein. In support of a link between plasma acylcarnitine levels and fatty acid oxidation in the hepatic tissue, the AC2/AC16 ratio positively correlated with mitochondrial fatty acid oxidation (Figure 6G). Fatty acid oxidation occurs in the mitochondria and peroxisomes, where peroxisomes preferentially oxidize longer chain fatty acids, whereas the mitochondria have higher specificity for shorter chain fatty acids [83]. Unlike in mitochondria, β-oxidation in peroxisomes does not depend on L-carnitine. Related to this, it was of interest that no change in ACOX activity was found in male rats fed CPH and chicken oil (Figure 6D). As L-carnitines and its metabolites were significantly increased both by whole chicken protein and CPH, compared to a casein control, some of the observed metabolic effects could be due to factors related to the protein source released during digestion in the stomach, such as amino acid composition, rather than specific bioactive peptides in CPH. Further studies are needed to elucidate this.

In male mice fed an obesogenic high-fat/high-sucrose diet associated with insulin resistance and a pro-inflammatory state, a 50% CPH diet prevented high plasma cytokine levels compared to a casein diet [23]. In the present study, a moderate high-fat diet was used, with less likelihood of obesity-associated inflammation. As a result, inflammatory levels were low in all groups, and plasma cytokines were not significantly lower in CPH-fed rats compared to casein control fed male rats. Notably, the diet with chicken fillet tended to increase inflammation compared to casein, and this was not observed in the CPH groups. Table 6 provides a summary of the findings, and includes the metabolic effects observed with chicken protein, CPH, CO, and combined CPH and CO.

The limitations of the current study are the relatively high number of groups, which can increase the chance of random significant findings, but also make it challenging to perform robust statistical analysis without losing biologically important effects. To alleviate this, we pre-designed sub-experiments for statistical analysis, and used posthoc tests that were stringent. The advantage of the study is the inclusion of relevant unhydrolysed protein controls, and that the observed dose effects strengthen the findings.

## 5. Conclusions

In conclusion, we found that chicken and pork protein diets did not influence the adipose tissue mass or plasma lipids and hepatic fatty acid metabolism compared to casein diets in rats. However, a crude chicken protein hydrolysate containing potentially bioactive peptides lowered cholesterol, particularly in small lipoprotein particles. Moreover, the hepatic mitochondrial fatty acid oxidation efficiency was increased by the combination of CPH and chicken oil, as also reflected in plasma by an increased AC2/AC16 ratio that correlated positively to hepatic palmitoyl-CoA oxidation. Altogether, the findings indicate additive bioactivity of the combination diet on parameters involved in fatty acid oxidation, resulting in a lower weight gain. To further determine mechanisms of action it will be necessary to identify the bioactive peptides. The potential health beneficial effects of CPH in humans should be investigated in both men and women.

## Figures and Tables

**Figure 1 nutrients-14-05364-f001:**
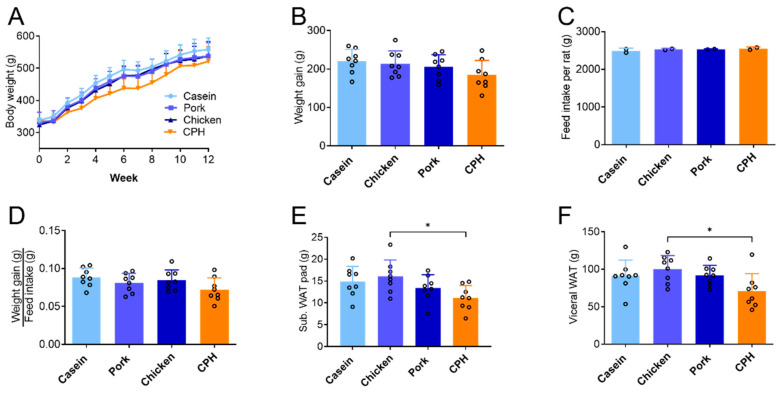
Body weight and feed intake in Wistar rats fed high-fat diets with different protein sources for 12 weeks, either casein, chicken meat, pork meat, or hydrolyzed chicken protein (CPH). (**A**) Body weight per rat, mean +/− SD are shown (*n* = 8). (**B**) Total weight gain per rat during the study, mean +/− SD are shown (*n* = 8). (**C**) Total feed intake per rat measured throughout the study. Mean values were based on average feed intake per rat per cage (*n* = 2). (**D**) Feed efficiency was based on total weight gain per rat divided by average feed intake per rat per cage (*n* = 2). (**E**) Subcutaneous WAT depot weights, mean +/− SD, are shown (*n* = 8). (**F**) Sum of weights of epididymal/gonodal WAT depots, perirenal WAT-depots and the mesenteric WAT depot, mean +/− SD are shown (*n* = 8). Individual values are indicated by circles Significant difference was determined by one-way ANOVA with Tukey’s multiple comparisons test between all diet groups (* *p* < 0.05).

**Figure 2 nutrients-14-05364-f002:**
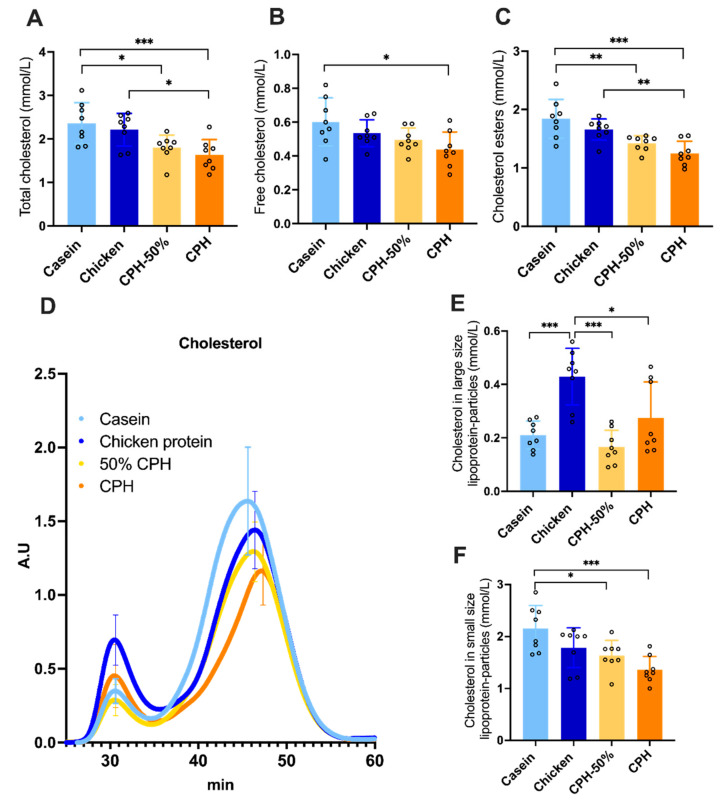
Dose effect on plasma cholesterol levels in Wistar rats fed high-fat diets with casein, chicken meat, or different amounts of chicken protein hydrolysate for 12 weeks. In the 50% CPH group, 50% of the protein source was casein. (**A**) Enzymatic measurement of total cholesterol and (**B**) free cholesterol, (**C**) calculated values of cholesterol esters from (**A**,**B**). (**D**) Chromatogram from size-exclusion chromatography analysis of cholesterol in lipoprotein particles. (**E**) Cholesterol concentration in large size and (**F**) smaller sized lipoprotein particles. Means values with standard deviation are given, and circles indicate individual values (*n* = 8). Significant difference in (**A**–**F**) was determined by one-way ANOVA with Tukey’s multiple comparisons test between all diet groups (* *p* < 0.05, ** *p* < 0.01, *** *p* < 0.001).

**Figure 3 nutrients-14-05364-f003:**
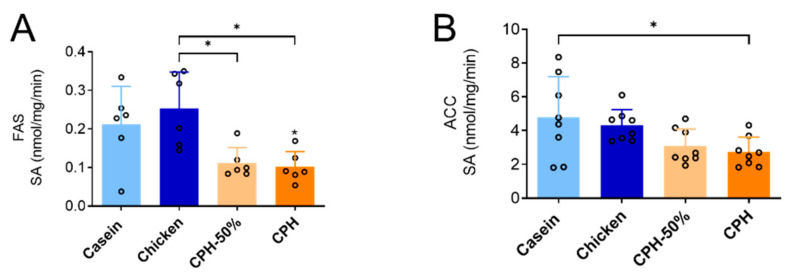
Dose effect on hepatic enzyme activity in Wistar rats fed high-fat diets with casein, chicken whole protein or different doses of chicken protein hydrolysate (CPH) for 12 weeks. In the 50% CPH group, 50% of the protein source was casein. (**A**) Fatty acid synthase (FAS) activity, and (**B**) acetyl-CoA carboxylase (ACC) activity was measured in frozen liver homogenates. Means values with standard deviation are given, and individual values are indicated by circles (*n* = 6–8). Significant difference was determined by one-way ANOVA with Tukey’s multiple comparisons test between all diet groups (* *p* < 0.05).

**Figure 4 nutrients-14-05364-f004:**
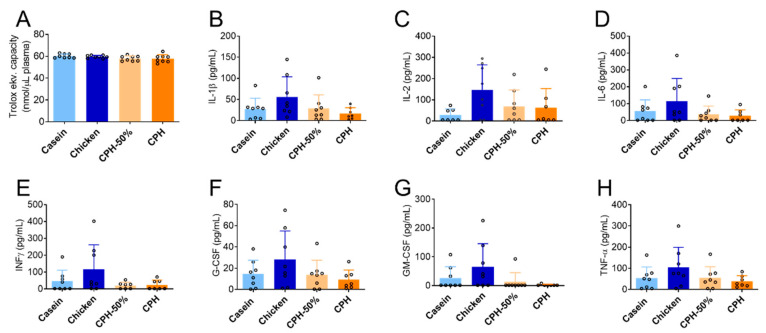
Dose effect of chicken protein hydrolysate (CPH) on plasma antioxidant and inflammatory parameters. (**A**) Plasma antioxidant capacity (AOC) and (**B**–**H**) cytokine and chemokine levels in Wistar rats fed high-fat diets with casein, chicken protein (meat) or different doses of CPH for 12 weeks. Means values with standard deviation are given, and individual values are indicated by circles (*n* = 6–8). Significant difference was determined by one-way ANOVA with Tukey’s multiple comparisons test between all diet groups (no significance at *p* < 0.05).

**Figure 5 nutrients-14-05364-f005:**
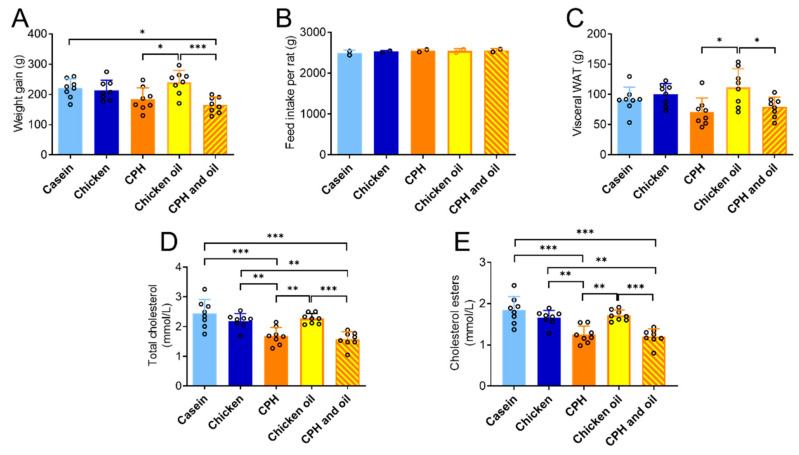
The effect of chicken oil, or a combination of chicken protein hydrolysate (CPH) and chicken oil on (**A**) weight gain, (**B**) total feed intake, (**C**) sum of visceral white adipose tissue (WAT) weights, (**D**) plasma total cholesterol and (**E**) cholesterol esters in Wistar rats. Means values with standard deviation are given, and individual values are indicated by circles (*n* = 7–8). One-way ANOVA with Tukey’s multiple comparisons test between all diet groups (* *p* < 0.05, ** *p* < 0.01, *** *p* < 0.001).

**Figure 6 nutrients-14-05364-f006:**
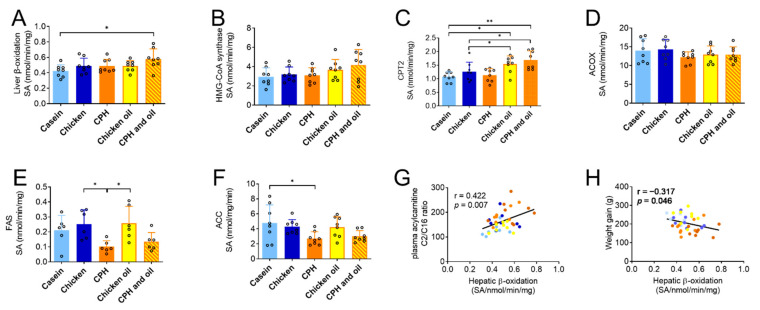
Hepatic enzyme activity in Wistar rats fed high-fat diets with casein, chicken protein (meat), chicken protein hydrolysate (CPH), chicken oil, or a combination of CPH and chicken oil for 12 weeks. (**A**) β-oxidation with or without malonyl CoA was analyzed in fresh samples, while the activity of (**B**) carnitine palmitoyltransferase II (CPT2), (**C**) HMG-CoA-Synthase, (**D**) peroxisomal acyl-conezyme A oxidase 1 (ACOX1), (**E**) fatty acid synthase (FAS), (**F**) acetyl-CoA carboxylase (ACC) was measured in frozen liver homogenates. Means values with standard deviation are given, and individual values are indicated by circles (*n* = 6–8). One-way ANOVA with Tukey’s multiple comparisons test between all diet groups was performed (* *p* < 0.05, ** *p* < 0.01). Pearson correlation coefficient (two-way) was calculated between the acetylcarnitine (AC2)/palmitoylcarnitine (AC16) ratio and hepatic β-oxidation (**G**), and between rat weight gain and hepatic β-oxidation (**H**). Light blue is control, dark blue is chicken protein, orange is CPH, yellow is chicken oil, and dark orange is CPH and chicken oil (**G**,**H**).

**Table 1 nutrients-14-05364-t001:** Nutrient composition in the diets calculated without water.

	Casein	Chicken	Pork	CPH-50%	CPH	CO	CPH and CO
Fat (47.5 E%)							
Soy oil (g)	24.0	24.0	24.0	24.0	24.0	24.0	24.0
Lard (g)	225.0	202.6	189.5	204.4	184.1	127.2	83.3
Chicken oil (g)	-	-	-	-	-	100.8	100.8
Fat from the protein source	3.0	25.4	38.5	23.6	43.9	3.0	43.9
Carbohydrate (32.4 E%)							
Cornstarch (g)	108.0	108.0	108.0	108.0	108.0	108.0	108.0
Maltodextrin (g)	158.6	158.6	158.6	158.6	158.6	158.6	158.6
Sucrose (g)	120	120	120	120	120	120	120
Fiber (g)	60	60	60	60	60	60	60
Protein (20.1 E%) ^1^							
Casein (g)	240	-	-	120	-	240	-
Pork protein (g)	-	-	240	-	-	-	-
Chicken protein (g)	-	240	-	-	-	-	-
CPH (g)	-	-	-	120	240	-	240
Micronutrients ^2^							
AIN-93G-MX mineral mix (g)	42.8	42.8	42.8	42.8	42.8	42.8	42.8
AIN-93-VX vitamin mix (g)	12.2	12.2	12.2	12.2	12.2	12.2	12.2
L-Cysteine (g)	3.67	3.67	3.67	3.67	3.67	3.67	3.67
Choline bitartrate (g)	2.5	2.5	2.5	2.5	2.5	2.5	2.5
tert-Butyl-hydroquinone (g)	0.0171	0.0171	0.0171	0.0171	0.0171	0.0171	0.0171
Total (g)	1000.0	1000.0	1000.0	1000.0	1000.0	1000.0	1000.0

Abbreviations: CPH, chicken protein hydrolysate-100% of the protein source; CPH-50%, 50% of the protein source; CO, chicken oil. ^1^ Protein content of the different protein sources: Casein—87%, chicken fillet—22.9%, pork fillet—21.7%, CPH—21.3%. The water content of the protein sources varied, and water was added to the diets accordingly. ^2^ Amounts based on AIN-93G adjusted for energy-dense diets.

**Table 2 nutrients-14-05364-t002:** Fatty acid composition of chicken oil.

Fatty Acid	%
14:0	0.82
14:1	0.21
15:0	0.14
16:0	22.5
16:1*n*-9	0.42
16:1*n*-7	5.28
17:0	0.23
18:0	6.64
18:1*n*-11 + 18:1*n*-9	39.0
18:1*n*-7	1.93
18:2*n*-6	19.4
18:3*n*-6	0.17
18:3*n*-3	1.57
18:4*n*-3	0.18
20:0	0.081
20:1*n*-11	0.14
20:1*n*-9	0.45
20:2*n*-6	0.18
20:3*n*-6	0.11
20:4*n*-6	0.22
20:3*n*-3	0.043
20:4*n*-3	0.019
20:5*n*-3	0.070
22:0	0.035
22:1*n*-11	0.037
22:1*n*-9	0.021
22:5*n*-3	0.073
24:0	0.003
22:6*n*-3	0.060
24:1*n*-9	0.013
Sum	100.0
Sum SFA	30.4
Sum MUFA	47.5
Sum PUFA	22.1
Sum omega-3 PUFA	2.0
Sum omega-6 PUFA	19.9
Sum EPA, DHA, DPA	0.2

**Table 3 nutrients-14-05364-t003:** Analysis of ash, fat, protein, water, energy, and amino acid composition, in high-fat diets with different protein or lipid sources.

	Casein	Chic.	Pork	CPH-50%	CPH	CO	CPH and CO
Ash (%)	2.2	2.3	2.2	2.6	2.9	2.1	2.9
Fat (%)	13.2	11.7	11.8	12.9	12.9	13.3	12.9
Protein (%) ^1^	13.1	12.6	13.2	14.6	13.3	13.1	14.1
Moisture (%)	46.0	48.7	47.2	47.5	45.8	46.2	45.4
Combustion value (Kilojoule/g	12.9	11.8	12.0	12.4	12.2	12.7	12.2
Aspartic acid (%) ^2^	7.63	9.16	9.92	7.63	6.49	7.56	6.56
Glutamic acid (%) ^2^	22.9	14.5	16.0	19.1	13.7	22.1	13.7
Hydroksyproline (%) ^2^	-	-	-	4.12	6.18	-	6.49
Serine (%) ^2^	6.03	3.97	4.35	4.89	3.36	5.88	3.44
Glycine (%) ^2^	1.98	4.35	4.81	9.16	12.21	1.91	13.74
Histidine (%) ^2^	2.98	2.82	4.05	2.44	1.76	2.90	1.76
Arginine (%) ^2^	3.74	6.03	6.72	5.50	5.95	3.51	6.49
Threonine (%) ^2^	4.58	4.27	4.96	3.82	2.75	4.35	2.90
Alanine (%) ^2^	3.21	5.27	5.80	5.65	6.64	3.05	6.87
Proline (%) ^2^	11.5	3.51	4.12	9.92	7.48	10.7	7.63
Tyrosine (%) ^2^	3.97	2.44	2.75	2.75	1.22	4.35	1.30
Valine (%) ^2^	6.87	5.04	5.57	4.81	2.75	6.49	2.98
Methionine (%) ^2^	2.98	2.67	2.98	2.52	1.76	2.90	1.91
Isoleucine (%) ^2^	5.65	4.96	5.42	4.05	2.29	5.34	2.52
Leucine (%) ^2^	10.69	8.40	9.16	7.63	4.81	9.92	5.19
Phenylalanine (%) ^2^	5.34	3.97	4.50	3.97	2.37	5.11	2.52
Lysine (%) ^2^	8.40	9.16	9.92	7.25	5.34	8.40	5.80
Cysteine/Cystine (%) ^2^	2.21	2.67	2.60	2.29	2.14	2.21	2.21
Trypthophane (%) ^2^	1.15	1.07	1.30	0.84	n.a.	1.15	n.a.

Abbreviations: CPH, chicken protein hydrolysate-100% of the protein source; CPH-50%, chicken protein hydrolysate-50% of the protein source; CO, chicken oil; n.a., not applicable. ^1^ Crude protein. Kjeldahl (*n* × 6.25). ^2^ Percent of total protein.

**Table 4 nutrients-14-05364-t004:** Plasma lipid levels in Wistar rats fed high-fat diets chicken fillet, chicken protein hydrolysate (CPH) or chicken oil for 12 weeks ^1^.

	Casein	Chicken	CPH	CO	CPH and CO
HDL cholesterol (mmol/L)	1.91 ± 0.32 ^a^	1.62 ± 0.22 ^a,b^	1.32 ± 0.22 ^b^	1.77 ± 0.20 ^a^	1.31 ± 0.24 ^b^
LDL cholesterol (mmol/L)	0.49 ± 0.09 ^a^	0.38 ± 0.10 ^a^	0.23 ± 0.09 ^b^	0.43 ± 0.13 ^a^	0.15 ± 0.07 ^b^
Triglycerides (mmol/L)	1.02 ± 0.28 ^a^	1.26 ± 0.42 ^a^	1.10 ± 0.32 ^a^	1.06 ± 0.35 ^a^	1.12 ± 0.44 ^a^
Phospholipids (mmol/L)	2.01 ± 0.39 ^a^	1.90 ± 0.26 ^a,b^	1.67 ± 0.27 ^a,b^	1.87 ± 0.18 ^a,b^	1.56 ± 0.23 ^b^
NEFA (mmol/L)	0.31 ± 0.08 ^a^	0.28 ± 0.08 ^a^	0.34 ± 0.11 ^a^	0.30 ± 0.07 ^a^	0.28 ± 0.04 ^a^

Abbreviations: CPH, chicken protein hydrolysate; CO, chicken oil; NEFA, non-esterified fatty acids. ^1^ Mean values ± standard deviations are shown (*n* = 8). Different superscript letters indicate significantly different values as determined by one-way ANOVA with Tukey’s post hoc test. *p* < 0.05.

**Table 5 nutrients-14-05364-t005:** Plasma levels of L-carnitine, L-carnitine precursors and acylcarnitines in rats fed high-fat diets with different protein or lipid sources ^1^.

	Casein	Chicken	CPH	CO	CPH and CO
Carnitine, µM	41.2 ± 6.5 ^a^	54.4 ± 4.0 ^b^	53.7 ± 8.3 ^b,c^	46.3 ± 5.2 ^a,b^	58.3 ± 9.2 ^c^
Acetylcarnitine (AC2), µM	17.3 ± 2.9 ^a^	23.49 ± 6.3 ^a,b^	28.4 ± 4.6 ^b,c^	19.3 ± 3.5 ^a^	31.5 ± 6.2 ^c^
Propionoylcarnitine (AC3), µM	0.56 ± 0.14 ^a^	0.86 ± 0.25 ^a,b^	0.90 ± 0.30 ^a,b^	0.68 ± 0.14 ^a^	1.12 ± 0.33 ^b^
Valerylcarnitine (AC5), µM	0.12 ± 0.02 ^a^	0.13 ± 0.03 ^a^	0.11 ± 0.03 ^a^	0.13 ± 0.03 ^a^	1.13 ± 0.04 ^a^
Octanoyl-canitine (AC8), µM	0.021 ± 0.003 ^a^	0.024 ± 0.004 ^a,b^	0.026 ± 0.004 ^a,b^	0.024 ± 0.007 ^a,b^	0.026 ± 0.002 ^b^
Lauroylcarnitine (AC12), µM	0.015 ± 0.002 ^a^	0.017 ± 0.002 ^a^	0.018 ± 0.002 ^a^	0.016 ± 0.002 ^a^	0.018 ± 0.001 ^a^
Myristoylcarnitine (AC14), M	0.026 ± 0.005 ^a^	0.028 ± 0.004 ^a^	0.030 ± 0.004 ^a^	0.026 ± 0.006 ^a^	0.028 ± 0.002 ^a^
Palmitoylcarnitine (AC16), µM	0.14 ± 0.02 ^a^	0.14 ± 0.02 ^a^	0.15 ± 0.01 ^a^	0.14 ± 0.02 ^a^	0.16 ± 0.01 ^a^
Total carnitine, µM ^2^	59.3 ± 7.5 ^a^	79.0 ± 8.0 ^b,c^	83.3 ± 11.0 ^b,c^	66.6 ± 8.0 ^a,b^	91.3 ± 14.2 ^c^
AC2/AC16	128.2 ± 23.4 ^a^	165.7 ± 33.2 ^a,b,c^	186.1 ± 33.6 ^b,c^	140.0 ± 19.1 ^a,b^	201.5 ± 45.5 ^c^
Trimethyllysine, µM	0.81 ± 0.13 ^a,b^	0.86 ± 0.10 ^a^	0.79 ± 0.05 ^a,b^	0.90 ± 0.06 ^a^	0.73 ± 0.07 ^b^
γ-butyrobetaine, µM	1.04 ± 0.16 ^a^	2.10 ± 0.28 ^b^	1.43 ± 0.27 ^c^	1.14 ± 0.13 ^a,c^	1.21 ± 0.18 ^c^
TMAO, µM	0.69 ± 0.13 ^a^	0.85 ± 0.22 ^a,c^	1.42 ± 0.57 ^b^	0.75 ± 0.25 ^a^	1.33 ± 0.49 ^b,c^
Betaine, µM	80.9 ± 10.5 ^a^	97.7 ± 12.1 ^a^	128.3 ± 11.0 ^b^	92.8 ± 20.8 ^a^	102.4 ± 17.0 ^a^
Choline, µM	12.1 ± 1.6 ^a^	12.5 ± 1.5 ^a^	13.2 ± 2.1 ^a^	12.9 ± 2.1 ^a^	13.6 ± 2.1 ^a^

Abbreviations: AC, acylcarnitines; CPH, chicken protein hydrolysate; CO, chicken oil; TMAO, trimethylamine oxide. ^1^ Mean values ± standard deviations are shown (*n* = 8). Different superscript letters indicate significantly different values as determined by one-way ANOVA with Tukey’s post hoc test. *p* < 0.05. ^2^ Sum of L-carnitine and all measured AC.

**Table 6 nutrients-14-05364-t006:** Summary of metabolic parameters affected by diets with chicken protein (CP), chicken protein hydrolysate (CPH) and/or chicken oil (CO) compared to a casein and lard control diet ^1^.

Metabolic Effect	CP	CPH	CO	CPH + CO
Plasma parameters:				
Total cholesterol conc.	=	↓	=	↓
Cytokines	↑	=	=	=
γ-butyrobetaine	↑	↑	=	=
L-carnitine	↑	↑	=	↑
Total carnitine	↑	↑	=	↑
AC2/AC16 ratio	↑	↑	=	↑
Betaine	=	↑	=	=
TMAO	=	↑	=	↑
Hepatic enzyme activity:				
Lipogenesis	=	↓	=	(↓)
β-oxidation	=	=	(↑)	↑
Anthropometry:				
Weight gain	=	=	=	↓
Adipose tissue weight	=	↓	=	↓

^1^ Arrows indicate direction of change compared to a casein and lard control diet. Green arrows indicate an additive effect of CPH and CO.

## Data Availability

The data presented in this study are available on request from the corresponding author. The data are not publicly available due to disclosed reasons.

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
