# Peer review of "Plasma Cholesterol- and Body Fat-Lowering Effects of Chicken Protein Hydrolysate and Oil in High-Fat Fed Male Wistar Rats"

_nutrients, 2022, doi:10.3390/nu14245364_

Round 1

Reviewer 1 Report

This manuscript is very interesting that showing beneficial effect of processed chicken rest, oil and protein hydrolysate. From the viewpoint from SDGs, this result might have important impact. But I think some revision and additional experiments are needed.

Table1 Why the column “Olive oil” exist?

Detail explanation about “corolaze PP” is needed. Is this enzymes for commercial use? Mixture of protease?

Identification, especially determination of amino acid sequence of the functional peptide is required. Purification is important for considering the effect of chicken protein hydrolysate.

It should also be verified with reference to the amino acid sequence of actin and other databases to determine from which site the excised peptide shows action. Determination of the origin of the functional peptide is also important issue.

Figure 4 According to the manuscript, Dunnett’s multiple comparison test was used for statistical analysis. This method can be used for comparison of treatment groups with “a single control”. To avoid multiple comparison problem other method should be chosen and Statistical analysis should not be repeated. Tukey's multiple comparison test should be good for the trial’s purpose.

Author Response

Dear reviewer no 1,

We thank you for your comments, and have tried to answer to the best of our ability. See specified answers below. The manuscript has been changed according to the comments, and we believe that the changes have improved the quality of the manuscript.

Sincerely,

Bodil Bjørndal

Reviewer no 1

Comments and Suggestions for Authors

This manuscript is very interesting that showing beneficial effect of processed chicken rest, oil and protein hydrolysate. From the viewpoint from SDGs, this result might have important impact. But I think some revision and additional experiments are needed.

Table1 Why the column “Olive oil” exist?

Reply: This control was excluded, and the column should have been removed. We apologize for this mistake.

Detail explanation about “corolaze PP” is needed. Is this enzymes for commercial use? Mixture of protease?

Reply: Corolase PP is a commercially available food grade enzyme from AB Enzymes GmbH. We apologize that the name of the distributer was missing, and the necessary information has been added to the methods section.

Identification, especially determination of amino acid sequence of the functional peptide is required. Purification is important for considering the effect of chicken protein hydrolysate.

Reply: We did not isolate the functional peptides in the crude chicken hydrolysate, although this is the focus of supplementary research projects. We agree that a complete mechanistic determination would require an isolated peptide, but it is also important to identify the properties of the crude hydrolysates as a dietary component. We here aim to determine of the CPH has beneficial metabolic properties compared to a range of unhydrolyzed protein sources. This will facilitate the use of rest raw materials for human consumption, and increase sustainability in the food industry.

It should also be verified with reference to the amino acid sequence of actin and other databases to determine from which site the excised peptide shows action. Determination of the origin of the functional peptide is also important issue.

Reply: We agree that this is scientifically interesting and should be performed in further research. However, as mentioned above this was not possible in this project. . The CPH will contain a number of short peptides with a range of potential bioactive properties. We have previously demonstrated that properties of CPH will vary depending on the enzymes used (https://pubmed.ncbi.nlm.nih.gov/30597839/), and that beta-oxidation is increased in a mouse obesity model. The following sentence was added to Discussion, 450-453: “As CPH produced by Corolase PP digestion has previously been shown to stimulate hepatic β-oxidation in mice [1], the increase in hepatic β-oxidation in the combination group could be caused by a dual effect generated by fatty acids in CO and bioactive peptides in CPH”
To further discuss the possibilities of effects by the source of protein, the following sentence was added to the discussion, line 500-502: “As L-carnitines and its metabolites were significantly increased both by whole chicken protein and CPH compared to a casein control, some of the observed metabolic effects could be due to factors related to the protein source, such as amino acid composition, rather than specific bioactive peptides in CPH. Further studies are needed to elucidate this.” We also added this sentence in the Conclusions, line 536-537: “To further determine mechanisms of action it will be necessary to identify the bioactive peptides.”

Figure 4 According to the manuscript, Dunnett’s multiple comparison test was used for statistical analysis. This method can be used for comparison of treatment groups with “a single control”. To avoid multiple comparison problem other method should be chosen and Statistical analysis should not be repeated. Tukey's multiple comparison test should be good for the trial’s purpose.

Reply: We agree with this comment, but there are several factors to consider when designing the animal experiment and statistical analysis. We made the choice to include more than one protein source as a comparison to the hydrolysate, as the use of only a casein control is often criticized. As casein is commonly used, we wished to include both this and chicken filet. When performing Tukey’s multiple comparison test, the number of comparisons makes the stringency higher with every added group, making it challenging to identify potentially biologically relevant effects. Thus, power analysis showed that the number of animals needed per group to identify differences using this test would not be ethically advisable. We have added a comment on this in the limitations of the study, line 520-526.

In addition, we noticed that minor changes to the text were needed, although not directly related to the comments, and made the following changes:

Removed sentence due to lack of relevance, line 46-47: “although recent meta-analyses have questioned the benefits of LC-n-3 on cardiovascular mortality and health [2, 3].”

Added sentence that was missing to add context to the previous sentence, line 74-75: “Importantly, TMAO has been suggested as a prognostic marker for non-communicable diseases [4].”

References:

  1. Aloysius T.A., Carvajal A.K., Slizyte R., Skorve J., Berge R.K., Bjorndal B. Chicken Protein Hydrolysates Have Anti-Inflammatory Effects on High-Fat Diet Induced Obesity in Mice. Medicines (Basel). 2018;6(1).
  2. Abdelhamid A.S., Brown T.J., Brainard J.S., Biswas P., Thorpe G.C., Moore H.J., et al. Omega-3 fatty acids for the primary and secondary prevention of cardiovascular disease. Cochrane Database Syst Rev. 2018;11:CD003177.
  3. Hall W.L. The future for long chain n-3 PUFA in the prevention of coronary heart disease: do we need to target non-fish-eaters? Proc Nutr Soc. 2017;76(3):408-18.
  4. Hoseini Tavassol Z., Ejtahed H.S., Larijani B., Hasani-Ranjbar S. Trimethylamine N-Oxide as a potential risk factor for non-communicable diseases: A systematic review. Endocr Metab Immune Disord Drug Targets. 2022.

Reviewer 2 Report

This manuscript investigated the effect of protein hydrolysate on plasma/body fat and suggested that CPH could lower blood total cholesterol levels. However, the possible mechanism behind the observed phenomenon was a lack of illustration.  Please see the comments below:

Title: the current title was a little bit long. Please consider shortening it, and what is Chicken Rest Raw Material? Is it the processing byproduct? please specify it in the material section.

Experimental design: why not set a group fed with chicken protein as a control to see whether hydrolysis plays a significant role?

Ln 73-85: the last paragraph of the introduction should state the objectives of the current study, and the conclusion of the study should be stated in the conclusion section.

Figure 6: the ordinates (x and y-axis) are too small, and it's hard for readers to see clearly.

Discussion section: A illustration figure/graph is encouraged to show the potential mechanism of the observed phenomenon.

Author Response

Dear reviewer no 2,

We thank you for your comments and have tried to answer to the best of our ability. See specified answers below. The manuscript has been changed according to the comments, and we believe that the changes have improved the manuscript.

Sincerely,

Bodil Bjørndal

Reviewer no 2

Comments and Suggestions for Authors

This manuscript investigated the effect of protein hydrolysate on plasma/body fat and suggested that CPH could lower blood total cholesterol levels. However, the possible mechanism behind the observed phenomenon was a lack of illustration.  Please see the comments below:

Title: the current title was a little bit long. Please consider shortening it, and what is Chicken Rest Raw Material? Is it the processing byproduct? please specify it in the material section.

Reply: We thank the reviewer for these helpful comments, and have shortened the title to: “Plasma cholesterol- and body fat-lowering effects of chicken protein hydrolysate and oil in high-fat fed male Wistar rats”

We have also expanded the description of the rest raw material in the methods section, and specified that this is the meat part of rest raw materials, line 120-121: “Rest raw materials  was generated by mechanical deboning of the remaining chicken meat after fileting (Nortura AS Hærland, Eidsberg, Norway), and was used for the hydrolysis experiments”

Experimental design: why not set a group fed with chicken protein as a control to see whether hydrolysis plays a significant role?

Reply: Yes, we agree that this is an important control, and as mentioned to reviewer no 1, we found it important to include both an unhydrolyzed casein and chicken protein control. We decided to use minced chicken filet as a control, as this is the most natural form of chicken protein, and we adjusted for different levels of water and oil in the different protein sources. This is described in Methods.

Ln 73-85: the last paragraph of the introduction should state the objectives of the current study, and the conclusion of the study should be stated in the conclusion section.

Reply:  We agree and have specified the objectives of the study more clearly (line 84) and removed the conclusions from the end of the introduction.

Figure 6: the ordinates (x and y-axis) are too small, and it's hard for readers to see clearly.

Reply: All figures have been altered with larger font on the axis.

Discussion section: A illustration figure/graph is encouraged to show the potential mechanism of the observed phenomenon

Reply: We have added a table summarizing the potential mechanisms (Table 6), and we hope this will improve the readability of the discussion.

In addition, we noticed that minor changes to the text were needed, although not directly related to the comments, and made the following changes:

Removed sentence due to lack of relevance, line 46-47: “although recent meta-analyses have questioned the benefits of LC-n-3 on cardiovascular mortality and health [1, 2].”

Added sentence that was missing to add context to the previous sentence, line 74-75: “Importantly, TMAO has been suggested as a prognostic marker for non-communicable diseases [3].”

References:

  1. Abdelhamid A.S., Brown T.J., Brainard J.S., Biswas P., Thorpe G.C., Moore H.J., et al. Omega-3 fatty acids for the primary and secondary prevention of cardiovascular disease. Cochrane Database Syst Rev. 2018;11:CD003177.
  2. Hall W.L. The future for long chain n-3 PUFA in the prevention of coronary heart disease: do we need to target non-fish-eaters? Proc Nutr Soc. 2017;76(3):408-18.
  3. Hoseini Tavassol Z., Ejtahed H.S., Larijani B., Hasani-Ranjbar S. Trimethylamine N-Oxide as a potential risk factor for non-communicable diseases: A systematic review. Endocr Metab Immune Disord Drug Targets. 2022.

Reviewer 3 Report

The proposed manuscript is formally excellent and demonstrates large number of methods used in experiments. The effects of protein hydrolysate are not strikingly novel, but chicken oil is an interesting matter. I suppose it is not as delecious as goose fat, but can you add more info about technology of its preparation, palatability, maybe some physical properties.

Author Response

Dear reviewer no 3,

We thank you for your positive comments and feedback to our manuscript. See specified answers below. The manuscript has been changed according to the comments, and we believe that the changes have improved the manuscript.

Sincerely,

Bodil Bjørndal

Reviewer no 3

Comments and Suggestions for Authors

The proposed manuscript is formally excellent and demonstrates large number of methods used in experiments. The effects of protein hydrolysate are not strikingly novel, but chicken oil is an interesting matter. I suppose it is not as delecious as goose fat, but can you add more info about technology of its preparation, palatability, maybe some physical properties.

Reply: Thank you so much for the feedback. We here aim to determine of the CPH has beneficial metabolic properties compared to a range of unhydrolyzed protein sources. This will facilitate the use of rest raw materials for human consumption, and increase sustainability in the food industry. We agree that palatability is an important part of this, but was unfortunately not a focus of our study – a part from noticing that the rats enjoyed the feed. We have added some further description on preparation in the methods section.

Line 119-120: “Rest raw materials  was generated by mechanical deboning of the remaining chicken meat after fileting (Nortura AS Hærland, Eidsberg, Norway), and was used for the hydrolysis experiments.”

Line 133: “, and all diets had similar palatability based on the feed intake of the rats”

Round 2

Reviewer 1 Report

If carnitine is considered as a functional substance as you added in modified manuscript, it might also be cleaved from the protein during the digestion in stomach. Additional experiment is required to show that the amount of carnitine is not increased by pepsin/pancreatin digestion of casein and chicken protein

Description about unusual statistical analyses in the limitation is sincere. However, I cannot judge whether such unusual statistical analyses are worthy of publication, so I leave it to the editors to decide.

Other parts were well corrected.

Author Response

Reply to reviewer no. 1, second revision

Thank you for the thorough work, and your helpful additional comments. We hope we have been able to answer your comments satisfactory.

First comment

If carnitine is considered as a functional substance as you added in modified manuscript, it might also be cleaved from the protein during the digestion in stomach. Additional experiment is required to show that the amount of carnitine is not increased by pepsin/pancreatin digestion of casein and chicken protein

Reply

We agree that the increase in carnitine could be directly linked to the chicken protein source, and substances that are released in the stomach. Additional experiments on this will require new animal studies. We have modified the discussion to specify this more clearly (line 495-496):

“As L-carnitines and its metabolites were significantly increased both by whole chicken protein and CPH compared to a casein control, some of the observed metabolic effects could be due to factors related to the protein source and released during the digestion in the stomach, such as amino acid composition, rather than specific bioactive peptides in CPH. Further studies are needed to elucidate this.”

Second comment

Description about unusual statistical analyses in the limitation is sincere. However, I cannot judge whether such unusual statistical analyses are worthy of publication, so I leave it to the editors to decide.

Reply

We thank the reviewer for suggesting that we reconsider the statistics. We originally ment that Dunnet’s test would be the best choice for the research question, but we do not wish to present statistics that are faulty or difficult to understand. We got a “second opinion” by a statistician, and although some considerations must be taken during animal studies, he provides some arguments that Tukey’s test is a better choice, despite the loss of potentially biologically relevant findings. We have altered all statistics in the revised version, see track changes.

I include the comments from the statistician, which I found quite helpful:

“Based on the provided information, here are my thoughts about the query you asked:

  • In my view, the Dunnett’s test for multiple comparisons is definitely useful when one wishes to compare experimental conditions with a control group. The power of this test is higher because the number of compared groups is reduced compared to the ‘all pairwise comparison’ tests like, e.g., the Tukey’s test. On top of that, Dunnett’s method is capable of ‘two-tailed’ or ‘one-tailed’ testing, which often helps if one has ‘directional’ hypotheses (i.e., there is association between … versus there is a positive/negative association between …).
  • However, the Dunnett’s test is not that well known. As a default, most of the researchers select the Tukey test or Tukey HSD test for the post-hoc multiple comparisons. On top of that, Dunnett’s method is useful when the researcher wishes to test two or more experimental groups against a single control group (McHugh, 2011). As in your case, you have two control groups, then your obtained p-values (from running Dunnett’s test twice on the same data) are not entirely correct (i.e., they seem to be adjusted for ‘within’ a single Dunnett’s test, but not ‘between’ the two runs of Dunnett’s tests – if that makes sense). Of course, it is difficult to say how large this misspecification is. It might be trivial. Some researchers argue that no adjustments are needed for multiple comparisons at all (e.g., Rothman (1990) or Gelman et al. (2012)). Be that as it may, if you applied Tukey’s test or any other all pairwise comparison test (for a neat overview, see for example Midway et al., (2020)), you would avoid the aforesaid problem, but at the same time the statistical power for detecting significant differences between treatment and control groups would go down as you compare more groups (that is, you also compare treatment groups with each other).  
  • Taken together, in my opinion the use of Dunnett’s test twice on the same data is not ‘wrong’, but it may lead to some sort of imprecision in obtained p-values. Again, the size of this imprecision is difficult to state. Maybe if you added some effect size measures and confidence intervals (CI) when running Dunnett’s tests then it would be a bit easier for reviewer/reader to make sense of results. This of course will not solve the described ‘p-value imprecision’ issue, but will definitely improve the interpretation of results.

Hope that was helpful at least a little bit. Used references are:

  • McHugh, M. L. (2011). Multiple comparison analysis testing in ANOVA. Biochemia medica, 21(3), 203-209.
  • Rothman, K. J. (1990). No adjustments are needed for multiple comparisons. Epidemiology, 43-46.  
  • Gelman, A., Hill, J., & Yajima, M. (2012). Why we (usually) don't have to worry about multiple comparisons. Journal of research on educational effectiveness, 5(2), 189-211.
  • Midway, S., Robertson, M., Flinn, S., & Kaller, M. (2020). Comparing multiple comparisons: practical guidance for choosing the best multiple comparisons test. PeerJ, 8, e10387.